DATA RELEASE

# Annotation of Hox cluster and Hox cofactor genes in the Asian citrus psyllid, *Diaphorina citri*, reveals novel features

Teresa D. Shippy[1], Prashant S. Hosmani[2], Mirella Flores-Gonzalez[2], Lukas A. Mueller[2], Wayne B. Hunter[3], Susan J. Brown[1], Tom D'Elia[4] and Surya Saha[2,5],*

1   Division of Biology, Kansas State University, Manhattan, KS 66506, USA
2   Boyce Thompson Institute, Ithaca, NY 14853, USA
3   USDA-ARS, U.S. Horticultural Research Laboratory, Fort Pierce, FL 34945, USA
4   Indian River State College, Fort Pierce, FL 34981, USA
5   Animal and Comparative Biomedical Sciences, University of Arizona, Tucson, AZ 85721, USA

## ABSTRACT

Hox genes and their cofactors are essential developmental genes specifying regional identity in animals. Hox genes have a conserved arrangement in clusters in the same order in which they specify identity along the anterior–posterior axis. A few insect species have breaks in the cluster, but these are exceptions. We annotated the 10 Hox genes of the Asian citrus psyllid *Diaphorina citri*, and found a split in its Hox cluster between the *Deformed* and *Sex combs reduced* genes – the first time a break at this position has been observed in an insect Hox cluster. We also annotated *D. citri* orthologs of the Hox cofactor genes *homothorax, PKNOX* and *extradenticle* and found an additional copy of *extradenticle* in *D. citri* that appears to be a retrogene. Expression data and sequence conservation suggest that the *extradenticle* retrogene may have retained the original *extradenticle* function and allowed divergence of the parental *extradenticle* gene.

**Submitted:**   09 September 2021

\* Corresponding author. E-mail: suryasaha@cornell.edu

Preprint submitted at https://doi.org/10.1101/2021.10.09.463765

Included in the series: *Asian citrus psyllid community annotation* (https://doi.org/10.46471/GIGABYTE_SERIES_0001)

Subjects   Genetics and Genomics, Animal Genetics, Bioinformatics

## DATA DESCRIPTION

### Background

Hox genes encode transcription factors best known for their role in specifying regional identity along the anterior–posterior (A–P) axis [1]. They were first identified in the fruit fly, *Drosophila melanogaster* [2–4], but are conserved throughout the Metazoa [1]. A phenomenon known as spatial collinearity, in which Hox genes cluster in an arrangement that parallels their function along the A–P axis, is also widely conserved [5, 6].

Researchers investigating the reasons for this conserved arrangement have discovered several constraints that could be keeping Hox clusters intact. First, the Hox clusters of vertebrates show temporal collinearity (the sequential activation of Hox genes from the 3′ end to the 5′ end of the cluster), which may be regulated by progressive opening of the chromatin to an active state [7–9]. Therefore, the need to maintain the timing of Hox gene expression during development could drive Hox cluster conservation. There is also

evidence that the presence of shared regulatory elements is a factor in the conservation of Hox clusters. In vertebrates, these include enhancers shared by two Hox genes [10–12] and global enhancer elements outside the cluster that regulate multiple Hox genes [6, 13, 14]. Others have suggested that the conservation of collinearity may stem from the need to minimize the number of boundaries between active and inactive chromatin to avoid inappropriate activation of genes [15].

In insects, the constraints on Hox clusters appear to be more relaxed than those affecting vertebrate clusters. The most recent common ancestor of insects is believed to have had 10 Hox genes: *labial* (*lab*), *proboscipedia* (*pb*), *zerknüllt* (*zen*), *Deformed* (*Dfd*), *Sex combs reduced* (*Scr*), *fushi tarazu* (*ftz*), *Antennapedia* (*Antp*), *Ultrabithorax* (*Ubx*), *abdominal-A* (*abd-A*) and *Abdominal-B* (*Abd-B*). Ancestrally, these genes were located in a single cluster and all transcribed on the same strand [16–18]. However, splits of the cluster have occurred in several insect lineages, including three independent splits in *Drosophila* species [19, 20] and one in the silk moth, *Bombyx mori* [21]. Inversions affecting the transcriptional direction of one or a few genes are also relatively common [18, 20]. Insect Hox clusters are larger than those of vertebrates, with longer genes (particularly the introns), and more intervening sequence between genes [18]. Unlike in vertebrates, temporal collinearity and global control elements do not seem to be a factor. Studies have suggested that the main constraint keeping insect Hox genes clustered is the presence of large regulatory regions between genes, which limits the number of places the cluster can be broken without affecting gene function [20]. There are also four microRNA (miRNA) genes conserved within arthropod Hox clusters [18]. At least some of these miRNAs target and repress translation of Hox mRNAs, particularly in the ventral nerve cord [22].

Another group of genes whose functions are closely tied to the Hox genes are the members of the PBX and MEINOX families of TALE class homeodomain proteins [23]. These proteins are best known as cofactors to Hox proteins, although they have Hox-independent functions as well [24–26]. The PBX and MEINOX proteins are ancient. They are conserved throughout the animal kingdom, and a related protein is thought to have been present in the common ancestor of plants and animals [27]. Most insects have one member of the PBX family and two members of the MEINOX family [28, 29]. The MEINOX genes consist of one member of the MEIS family and one member of the PREP/PKNOX family. In *Drosophila*, where the PBX and MEINOX proteins are best studied, the PREP/PKNOX ortholog has been lost, leaving only the PBX gene *extradenticle* (*exd*) [30] and the MEIS gene *homothorax* (*hth*) [31].

## Context

As part of a community gene annotation project, we are curating genes implicated in development, immune function and metabolism from the genome of the Asian citrus psyllid, *Diaphorina citri* (NCBI:txid121845) [32]. This hemipteran insect pest carries the bacterium *Candidatus* Liberibacter asiaticus (*C*Las), which causes Huanglongbing (citrus greening disease) and is a serious threat to citrus production worldwide. The Hox genes and their cofactors are potential targets for RNA interference (RNAi)-based pest control methods because of their essential role in development. Hox gene sequences have been identified previously in several hemipterans, including the milkweed bug *Oncopeltus fasciatus* and the pea aphid *Acyrthosiphon pisum*. These insects seem to have the full complement of Hox genes [29, 33–38]. Based on knowledge of other insects, hemipteran Hox genes are probably

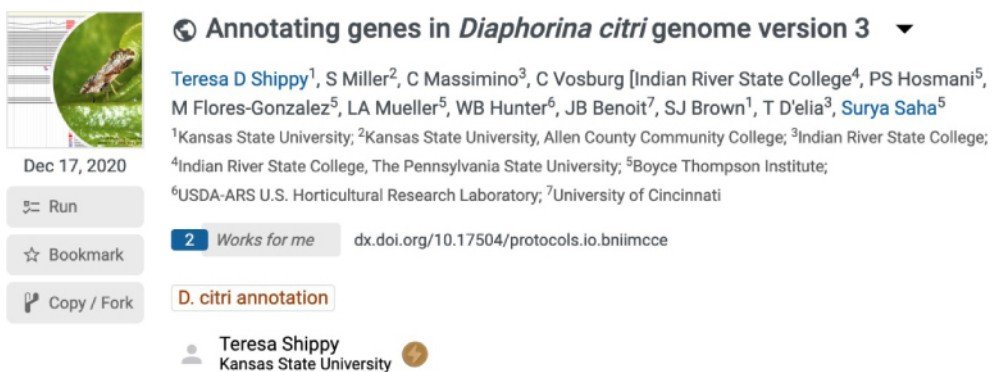

**Figure 1.** Protocol for psyllid genome community curation [42]. https://www.protocols.io/widgets/doi?uri=dx.doi.org/10.17504/protocols.io.bniimcce

clustered, but so far only a few examples of gene linkage have been reported owing to the fragmented condition of available genome assemblies. To date, linkage has only been demonstrated for a few pairs of genes: *zen – pb* and *ftz – Scr* in the pea aphid [29], and *zen – Dfd* in *Oncopeltus* [38].

Hemipteran Hox gene function has been best studied in *Oncopeltus*. RNAi experiments indicate that *Oncopeltus* Hox genes have functions broadly similar to their orthologs in holometabolous insects such as *Drosophila* and *Tribolium*, although there are many small differences in regulation and function [33, 34]. *Oncopeltus* studies show that, with most Hox genes, RNAi causes high levels of lethality [33, 34]. One exception is RNAi with *pb*: this produces viable nymphs with abnormal mouthpart morphology, so they cannot feed [33]. In the brown planthopper *Nilaparvata lugens*, RNAi with *zen* caused high levels of embryonic death and prevented normal development [39]. These results suggest that Hox genes could be targets for the development of RNAi-based pest management products in hemipteran pests such as *D. citri*. While specific regions of Hox proteins are highly conserved throughout the animal kingdom, their nucleotide sequences differ sufficiently to provide many regions for the design of highly specific RNAi-based products.

## METHODS

*D. citri* genes were identified by BLAST (NCBI BLAST, RRID:SCR_004870) analysis of *D. citri* sequences [40] with orthologs from *D. melanogaster* and *Tribolium castaneum*. Accession numbers are provided in Table 1. The only exception was *zen*, which was located by using the Apollo BLAT function to search the expected region of the Hox cluster for additional homeodomain-encoding sequences. Reciprocal BLAST against insect proteins was used to confirm the identity of all *D. citri* orthologs and to assess their completeness. Genes were manually annotated in Apollo 2.1.0 (Apollo, RRID:SCR_001936) [41] using RNA-seq reads, Iso-seq transcripts and *de novo*-assembled transcripts [40] as evidence of gene structure. A more detailed description of the annotation workflow, including an explanation of the use of reciprocal BLAST to identify orthologs, is available via protocols.io (Figure 1) [42]. Multiple alignments and phylogenetic analysis were performed with MEGA X (MEGA Software, RRID:SCR_000667) [43]. Orthologs used in phylogenetic analysis are listed in Table 1.

**Table 1.** Orthologs used in phylogenetic analysis and multiple sequence alignments.

| Species | Accession | Name in NCBI | Name in tree |
|---|---|---|---|
| *Acyrthosiphon pisum* | XP_008182670.1 | Homeobox protein extradenticle isoform X1 | Ap Exd |
| *Nilaparvata lugens* | XP_039288084.1 | Homeobox protein extradenticle | Nl Exd |
| *Bemisia tabaci* | XP_018913868.1 | Homeobox protein extradenticle isoform X2 | Bt Exd |
| *Apis mellifera* | XP_006558038.2 | Homeobox protein extradenticle isoform X4 | Am Exd |
| *Tribolium castaneum* | NP_001034501.1 | Extradenticle | Tc Exd |
| *Drosophila melanogaster* | NP_001259592.1 | Extradenticle, isoform D | Dm Exd |
| *Bombyx mori* | XP_004928674.1 | Homeobox protein extradenticle isoform X4 | Bm Exd |
| *Pachypsylla venusta* | ScZCZ4B_2870.g10167.t1 | | Pv Exd |
| *Diaphorina citri* | Dcitr08g10400.1.1 | | Dc Exd |
| *Pachypsylla venusta* | ScZCZ4B_298.g6815.t1 | | Pv Exd-r |
| *Diaphorina citri* | Dcitr01g03700.1.1 | | Dc Exd-r |
| *Homo sapiens* | XP_016856884.1 | Pre-B-cell leukemia transcription factor 1 isoform X4 | Hs PBC1 |
| *Nilaparvata lugens* | XP_039298141.1 | Homeobox protein Meis1 | Nl PKNOX |
| *Acyrthosiphon pisum* | XP_008181007.1 | LOW QUALITY PROTEIN: homeobox protein PKNOX2 | Ap PKNOX |
| *Tribolium castaneum* | XP_970138.1 | PREDICTED: homeobox protein PKNOX2 isoform X2 | Tc PKNOX |
| *Apis mellifera* | XP_001120618.1 | Homeobox protein PKNOX2 isoform X1 | Am PKNOX |
| *Bombyx mori* | XP_037875077.1 | Homeobox protein PKNOX2 isoform X1 | Bm PKNOX |
| *Homo sapiens* | BAB83665.1 | PKNOX2 | Hs PKNOX2 |
| *Diaphorina citri* | MCOT13578.2.CT | | Dc PKNOX |
| *Nilaparvata lugens* | XP_039282637.1 | Homeobox protein homothorax isoform X12 | Nl Hth |
| *Tribolium castaneum* | NP_001034489.1 | Homothorax | Tc Hth |
| *Apis mellifera* | XP_026296417.1 | Homeobox protein homothorax isoform X2 | Am Hth |
| *Drosophila melanogaster* | NP_476578.3 | Homothorax, isoform A | Dm Hth |
| *Bombyx mori* | NP_001296493.1 | Homeobox protein homothorax-like | Bm Hth |
| *Acyrthosiphon pisum* | XP_001951115.2 | Homeobox protein homothorax isoform X2 | Ap Hth |
| *Homo sapiens* | NP_002390.1 | Homeobox protein Meis2 isoform f | Hs Meis2 |
| *Diaphorina citri* | Dcitr03g04000.1.1 | | Dc Hth |

All orthologs used in phylogenetic analysis and multiple sequence alignment are listed with species name, accession number, full name from the National Center for Biotechnology Information (NCBI) and abbreviated name used in trees. Dcitr and MCOT numbers refer to *D. citri* sequences that can be retrieved from the Citrus Greening web site [44].

Expression counts for annotated genes were downloaded from the Citrus Greening Expression Network [40]. Heatmaps were produced with the pheatmap package (pheatmap, RRID:SCR_016418) in R (R Project for Statistical Computing, RRID:SCR_001905). Bar charts were created using Microsoft Excel (Microsoft Excel, RRID:SCR_016137).

To search for orthologs of *exd-r* in hemipteran genomes, we used the *D. citri* Exd-r protein as a query to BLAST the assemblies of *Pachypsylla venusta* (GCA_012654025.1), *Bemisia tabaci* (GCA_004919745.1 and GCA_003994315.1), *Trialeurodes vaporariorum* (GCA_011764245.1), and *A. pisum* (GCA_005508785.1).

**Table 2.** Gene counts in selected insect species.

| Hox genes | Dm | Am | Tc | Ap | Dc |
|---|---|---|---|---|---|
| *labial* | 1 | 1 | 1 | 1 | 1 |
| *proboscipedia* | 1 | 1 | 1 | 1 | 1 |
| *zerknüllt* (*Hox 3*) | 3 | 1 | 2 | 1 | 1 |
| *Deformed* | 1 | 1 | 1 | 1 | 1 |
| *Sex combs reduced* | 1 | 1 | 1 | 1 | 1 |
| *fushi tarazu* | 1 | 1 | 1 | 1 | 1 |
| *Antennapedia* | 1 | 1 | 1 | 1 | 1 |
| *Ultrabithorax* | 1 | 1 | 1 | 1 | 1 |
| *abdominal-A* | 1 | 1 | 1 | 1 | 1 |
| *Abdominal-B* | 1 | 1 | 1 | 2 | 1 |
| **Hox cluster miRNAs** | **Dm** | **Am** | **Tc** | **Ap** | **Dc** |
| *miR-10* | 1 | 1 | 1 | 1 | 2* |
| *miR-993* | 1 | 1 | 1 | 1 | 2* |
| *miR-iab-4* | 1 | 1 | 1 | 1 | 3* |
| *miR-iab-8* | 1 | 1 | 1 | 1 | 3* |
| **Hox cofactors** | **Dm** | **Am** | **Tc** | **Ap** | **Dc** |
| *extradenticle* | 1 | 1 | 1 | 1 | 2 |
| *homothorax* | 1 | 1 | 1 | 1 | 1 |
| *PKNOX* | 0 | 1 | 1 | 1 | 1 |

Comparison of gene copy number for Hox genes and Hox cluster miRNAs in *Drosophila melanogaster* (*Dm*), *Apis mellifera* (*Am*), *Tribolium castaneum* (*Tc*), *Acyrthosiphon pisum (Ap)* and *Diaphorina citri* (*Dc*) [18, 29, 47–52].
* Indicates miRNA genes that appear as multiple copies in *D. citri* genome v3, but are likely to be single copy genes with false duplications due to local misassembly.

## DATA VALIDATION AND QUALITY CONTROL

### Hox genes

We identified single orthologs of each of the Hox genes in *D. citri* (Tables 2, 3). Notably, *lab* is not found in the *D. citri* v3.0 genome, but is found in the Maker, Cufflinks, Oases, Trinity (MCOT) transcriptome and was present in the *D. citri* v1.1 and v2.0 genome assemblies [32, 40, 45] in its expected position adjacent to *pb*. Thus, its absence in *D. citri* genome v3.0 seems to be caused by a local genome misassembly. We also observed additional local misassemblies resulting in false duplications of several Hox genes, including *zen, ftz, Scr* and *Antp* in both the *D. citri* v2.0 and v3.0 assemblies. This type of duplication, in which almost identical copies of a genome segment are assembled in tandem within a scaffold, is usually caused by the presence of allelic variants mistakenly assembled as separate genes [46]. This is fairly common in all versions of the *D. citri* genome because of the heterogeneity of the sequenced psyllids [32].

In *D. citri* genome v3.0, the Hox genes (except for *lab* as discussed above) are found in two clusters on chromosome 7, separated by about 6 megabase pairs (Mbp). *pb, zen* and *Dfd* are found in one cluster (*lab* was also part of this cluster in a previous assembly), while *Scr, ftz, Antp, Ubx, abd-A* and *Abd-B* are located in a separate cluster (Figure 2). This indicates that a split in the Hox cluster has occurred at some point in the lineage leading to *D. citri*. The break, which is between *Dfd* and *Scr*, is in a different position to that of any previously described split in insect Hox clusters. In *D. melanogaster*, the Hox cluster is split between *Antp* and *Ubx,* while breaks between *Ubx* and *abd-A* and between *lab* and *pb* have been reported in other *Drosophila* species [20]. To date, the only other break described in a non-Drosophilid insect Hox cluster is the separation of *lab* from the rest of the cluster in the silk moth, *B. mori* [21].

**Ancestral Arthropod Hox Cluster**

**D. citri Hox Cluster(s)**

**Figure 2.** Comparison of Hox gene clusters in *Diaphorina citri* with the putative ancestral arthropod Hox cluster. In the ancestral cluster, all 10 Hox genes were oriented in the same direction, and at least four microRNAs (miRNAs; black triangles) were found within the cluster [18]. In *D. citri*, the Hox genes are split between two clusters on the same chromosome. *lab* is missing from genome v3.0 because of apparent misassembly but was present in previous genome versions adjacent to *pb*. *D. citri pb* and *Dfd* appear to be oriented in the opposite direction to the ancestral orientation, but this could be the result of local misassembly. *D. citri* has all four of the conserved Hox cluster miRNAs in the expected positions. The additional copies of the miRNAs described in the article are not shown, since they are likely to be false duplications.

**Table 3.** Annotated *Diaphorina citri* orthologs of Hox cluster and Hox cofactor genes.

| Gene | *D. citri* OGS3 identifier | Complete gene model | Evidence supporting annotation | | | |
|---|---|---|---|---|---|---|
| | | | MCOT | Iso-seq | RNA-seq | Ortholog |
| *labial* [†] | | in v2 | X | | | X |
| *proboscipedia* | Dcitr07g11290.1.1 Dcitr07g11290.1.2 | X | MCOT18081.0.CO | X | X | X |
| *zerknüllt* | Dcitr07g11300.1.1 | X | MCOT17344.0.CT | | X | X |
| *Deformed* | Dcitr07g11310.1.1 | X | MCOT13111.0.CT | X | X | |
| *Sex combs reduced* | Dcitr07g04890.1.1 | X | MCOT12386.0.CT | | | X |
| *fushi tarazu* | Dcitr07g04910.1.1 | X | MCOT02202.0.CC | | | |
| *Antennapedia* | Dcitr07g04960.1.1 | X | MCOT16702.0.CC | X | | |
| *Ultrabithorax* | Dcitr07g05020.1.1 | X | MCOT21836.0.CT | X | | X |
| *abdominal-A* | Dcitr07g05110.1.1 | X | | | X | X |
| *Abdominal-B* | Dcitr07g05150.1.1 | X | MCOT02185.0.CT | X | | |
| *miR-10* | | X | | | | X |
| *miR-993* | | X | | | | X |
| *miR-iab-4* | | X | | | | X |
| *miR-iab-8* | | X | | | | X |
| *extradenticle* | Dcitr08g10400.1.1 | X | MCOT05038.1.CT | | X | |
| *extradenticle-retro* | Dcitr01g03720.1.1 | X | | | | X |
| *homothorax* | Dcitr03g04000.1.1 Dcitr03g04000.1.2 | X | MCOT17109.0.CT | X | X | X |
| *PKNOX* [†] | Dcitr00g14970.1.1 | | MCOT13578.2.CT MCOT13578.1.CT | X | X | X |

Manually annotated genes from *D. citri* have been assigned an official gene set 3 (OGS3) ID. For protein-coding genes, completeness refers to the presence of the full open reading frame. For microRNA (miRNA) genes, a complete gene model indicates that the entire hairpin-forming portion of the gene (the pre-miRNA) has been annotated. [†] Denotes genes that were present in *D. citri* genome v2.0, but are absent or less complete in *D. citri* genome v3.0 [32]. Evidence types include the Maker Cufflinks Oases Trinity (MCOT) transcriptome, high-quality Iso-seq transcripts (Iso-seq), RNA-seq reads (RNA-seq), and ortholog sequences (Ortholog). The use of these evidence types has been previously described [42].

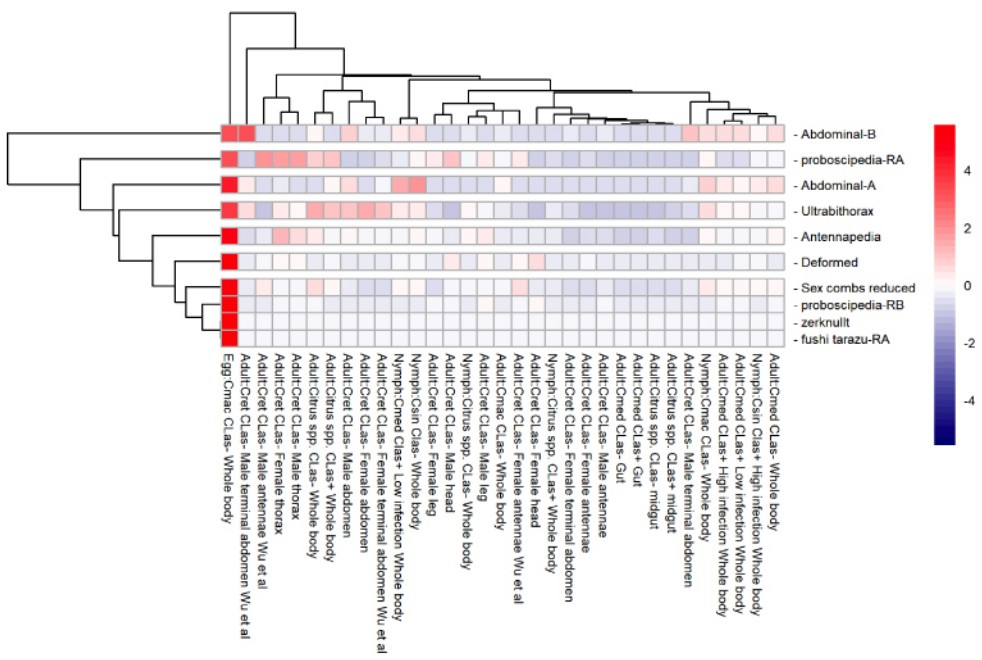

**Figure 3.** Expression of *Diaphorina citri* Hox genes. Heatmap of relative expression levels of Hox genes in RNA-seq samples of various developmental stages, tissues, host plants and *C*Las infection states ([53–57] and NCBI BioProjects PRJNA448935 and PRJNA609978). Host plants are *Citrus macrophylla* (Cmac), *Citrus reticulata* (Cret), *Citrus* spp., *Citrus medica* (Cmed), and *Citrus sinensis* (Csin). The heatmap was scaled by row to allow comparison of expression in different samples for each gene. All 10 genes showed highest expression in eggs.

In the current *D. citri* genome assembly, we find that *pb* and *Dfd* are oriented in the opposite direction to that which would be expected (Figure 2). It is possible that these inversions represent real rearrangements of the Hox cluster but, given the number of local misassemblies affecting the Hox cluster region, and the *D. citri* genome in general, we cannot be certain.

Intact insect Hox clusters range in size from 0.71 Mbp in *T. castaneum* to 1.22 Mbp in *Anopheles gambiae* [18]. As a comparison, we calculated the size of the two *D. citri* clusters. Since *lab* is missing in the *D. citri* v3.0 genome, we used a previous genome assembly (v2) [32] to determine that the portion of the cluster containing *lab, pb, zen, Dfd* and *miR-10* measures 231 kilobase pairs (Kbp). The part of the cluster containing the remaining Hox genes spans 2.17 Mbp in the *D. citri* v3.0 genome, but this length is almost certainly inflated by false duplications as described above.

We used publicly available RNA-seq data in the Citrus Greening Expression Network (CGEN) [40] to assess expression of *D. citri* Hox genes in various stages, tissues and *C*Las infection states. Expression of Hox genes is generally low in all samples (Table 4), with the highest expression in eggs (Figure 3). We observed no effect of *C*Las infection on Hox gene expression (Figure 3).

## Hox cluster miRNAs

There are several conserved miRNAs located within insect Hox clusters. *miR-10,* conserved in most animals, is found between the *Dfd* and *Scr* orthologs [58]. Three other miRNAs are widely conserved in arthropods [18]. *miR-993* is found between *Dfd* and *zen*, and the



**Table 4.** TPM expression values for *Diaphorina citri* Hox genes.

| Gene ID | Dcitr07g 11290.1.1 | Dcitr07g 11290.1.2 | Dcitr07g 11300.1.1 | Dcitr07g 11310.1.1 | Dcitr07g 04890.1.1 | Dcitr07g 04910.1.1 | Dcitr07g 04960.1.1 | Dcitr07g 05020.1.1 | Dcitr07g 05110.1.1 | Dcitr07g 05150.1.1 |
|---|---|---|---|---|---|---|---|---|---|---|
| Gene/transcript symbol | *pb*-RA | *pb*-RB | *zen* | *Dfd* | *Scr* | *ftz* | *Antp* | *Ubx* | *abd-A* | *Abd-B* |
| Egg *C. macrophylla* *C*Las− Whole body | 2.73 | 16.66 | 3.29 | 19.65 | 21.64 | 25.35 | 88.88 | 11.38 | 3.14 | 18.53 |
| Nymph *C. medica* *C*Las+ Low infection Whole body | 0.35 | 0 | 0 | 0.31 | 2.13 | 0 | 9.7 | 3.36 | 1.29 | 4.47 |
| Nymph *C. sinensis* *C*Las+ High infection Whole body | 0.47 | 0.01 | 0 | 0.35 | 2.41 | 0.02 | 8.85 | 2.09 | 0.6 | 3.55 |
| Nymph *C. sinensis* *C*Las− Whole body | 0.56 | 0.01 | 0 | 0.55 | 2.25 | 0 | 13.56 | 3.36 | 1.5 | 6.03 |
| Nymph *C. macrophylla* *C*Las− Whole body | 0.66 | 0 | 0 | 0.14 | 3.3 | 0 | 14.55 | 3.64 | 0.92 | 5.62 |
| Nymph *Citrus* spp. *C*Las− Whole body | 0.44 | 0 | 0 | 0 | 1.93 | 0 | 14.55 | 2.76 | 0 | 1.44 |
| Nymph *Citrus* spp. *C*Las+ Whole body | 0.22 | 0 | 0 | 0 | 1.12 | 0 | 6.6 | 1.38 | 0 | 0.13 |
| Adult *C. medica* *C*Las− Gut | 0.06 | 0 | 0 | 0 | 0 | 0 | 0.47 | 0.16 | 0.01 | 0 |
| Adult *C. medica* *C*Las+ Gut | 0.02 | 0 | 0 | 0 | 0.04 | 0 | 0.39 | 0.07 | 0 | 0.02 |
| Adult *C. medica* *C*Las+ High infection Whole body | 0.16 | 0.22 | 0 | 0.08 | 2.9 | 0 | 8.87 | 2.58 | 0.55 | 5.28 |
| Adult *C. medica* *C*Las+ Low infection Whole body | 0.15 | 0.22 | 0 | 0.32 | 2.35 | 0.06 | 10.34 | 2.66 | 0.42 | 5.56 |
| Adult *C. medica* *C*Las− Whole body | 0.54 | 0.14 | 0 | 0.18 | 2.45 | 0 | 12.03 | 2.28 | 0.66 | 5.63 |
| Adult *C. macrophylla* *C*Las− Whole body | 0.46 | 0 | 0 | 0 | 1.92 | 0 | 6.2 | 1.85 | 0.47 | 1.19 |
| Adult *Citrus* spp. *C*Las− Whole body | 1.08 | 0.34 | 0 | 0 | 3.86 | 0 | 17.89 | 6.25 | 0 | 3.32 |
| Adult *Citrus* spp. *C*Las+ Whole body | 1.2 | 0 | 0 | 0.56 | 2.58 | 0 | 10.99 | 5.02 | 0.41 | 0.82 |
| Adult *Citrus* spp. *C*Las− midgut | 0 | 0 | 0 | 0 | 0 | 0 | 1.4 | 0.13 | 0 | 0 |
| Adult *Citrus* spp. *C*Las+ midgut | 0 | 0 | 0 | 0 | 0.29 | 0 | 0.42 | 0.29 | 0 | 0.03 |
| Adult *C. reticulata* *C*Las− Female abdomen | 0.07 | 0 | 0 | 0 | 0 | 0 | 11.06 | 5.97 | 0 | 1.47 |
| Adult *C. reticulata* *C*Las− Female antennae | 0.2 | 0 | 0 | 0.7 | 0 | 0 | 1.85 | 0.14 | 0 | 0 |
| Adult *C. reticulata* *C*Las− Female head | 0 | 1.23 | 0 | 3.11 | 1.05 | 0 | 4.65 | 0.13 | 0 | 0 |
| Adult *C. reticulata* *C*Las− Female leg | 0.83 | 0.29 | 0 | 0.16 | 0 | 0 | 9.84 | 0.86 | 0 | 0 |
| Adult *C. reticulata* *C*Las− Female terminal abdomen | 0 | 0 | 0 | 0 | 0 | 0 | 0.25 | 0.87 | 0 | 1.37 |
| Adult *C. reticulata* *C*Las− Female thorax | 1.67 | 0.37 | 0 | 1.45 | 1.57 | 0 | 29.25 | 3.34 | 0.22 | 0 |
| Adult *C. reticulata* *C*Las− Male abdomen | 0.03 | 0 | 0 | 0 | 1.17 | 0 | 13.81 | 5.07 | 0.7 | 7.25 |
| Adult *C. reticulata* *C*Las− Male antennae | 0 | 0.39 | 0 | 0.22 | 0.71 | 0 | 2.07 | 0 | 0.05 | 1.06 |
| Adult *C. reticulata* *C*Las− Male head | 1.18 | 0.13 | 0 | 1.94 | 1.53 | 0 | 6.83 | 0 | 0 | 0.07 |
| Adult *C. reticulata* *C*Las− Male leg | 0.72 | 0.74 | 0 | 1.21 | 1.78 | 0 | 17.81 | 1.92 | 0 | 0.07 |
| Adult *C. reticulata* *C*Las− Male terminal abdomen | 0.06 | 0 | 0 | 0 | 0.79 | 0 | 0.61 | 1.04 | 0.16 | 7.97 |
| Adult *C. reticulata* *C*Las− Male thorax | 1.68 | 0.41 | 0 | 1.11 | 1.32 | 0 | 19 | 2.54 | 0 | 0 |
| Adult *C. reticulata* *C*Las− Female antennae [53] | 0.73 | 0.48 | 0 | 1.07 | 4.04 | 0 | 7.08 | 0.9 | 0 | 0.22 |
| Adult *C. reticulata* *C*Las− Female terminal abdomen [53] | 0.15 | 0 | 0 | 0 | 0.38 | 0 | 7.98 | 5.19 | 0.26 | 1.92 |
| Adult *C. reticulata* *C*Las− Male antennae [53] | 1.72 | 0 | 0 | 0.67 | 3.11 | 0 | 7.39 | 0.06 | 0 | 0.75 |
| Adult *C. reticulata* *C*Las− Male terminal abdomen [53] | 0 | 0 | 0 | 0 | 1.29 | 0 | 3.41 | 3.74 | 0.58 | 18.85 |

Expression levels of Hox genes used in Figure 3. Counts were obtained from publicly available datasets in CGEN [40] and are reported in transcripts per million (TPM). Developmental stage, citrus host, *C*Las infection status and source tissue for each sample are noted in the first column.

bidirectionally transcribed locus producing the *iab-4* and *iab-8* miRNAs is found between *abd-A* and *Abd-B* [59, 60]. We found multiple copies of each of the Hox cluster miRNAs in the *D. citri* v3.0 genome (Table 2). These additional copies are probably false duplications caused by local misassembly, particularly since the number often varies between different

versions of the assembly. In all but one case, the duplicate copies are adjacent to one another in the genome. The exception is *miR-993*, for which there is one copy on each side of *Dfd* in *D. citri* genome v3.0 (although there was only one copy in the expected position between *Dfd* and *zen* in genome v2.0). The apparent split in the *D. citri* Hox cluster is between *Scr* and *miR-10*, leaving *miR-10* adjacent to *Dfd*.

## MEINOX genes

MEINOX genes are broadly conserved cofactors of Hox genes. Like most insects, *D. citri* has two MEINOX genes (Tables 2, 3). One is orthologous to *hth* and the vertebrate *Meis* genes, while the other is orthologous to the *PREP/PKNOX* genes of insects and vertebrates (Figure 4). Therefore, we have named the two genes *hth* and *PKNOX*. The PKNOX locus is incomplete in *D. citri* genome v3.0, but the complete protein sequence can be deduced from *de novo*-assembled transcripts (Table 3).

Comparison of expression levels of *hth* and *PKNOX* in available *D. citri* RNA-seq datasets [40] shows a potentially interesting difference in expression between the two genes. *PKNOX* is expressed at higher levels in most adult samples than the *hth* isoform A (*hth-A*). Conversely, *hth-A* is expressed at higher levels than *PKNOX* in some, but not all, nymph samples (Figure 5, Table 5). Studies in vertebrates have found that the PKNOX/Prep and Hth/Meis family proteins not only have different DNA binding sites, but also preferentially bind to the regulatory regions of different types of genes. Meis tends to bind to genes involved in embryonic development, while Prep1 is more likely to bind to genes involved in housekeeping functions [61]. Thus, it is interesting that the *D. citri hth/Meis* ortholog seems to be up-regulated in nymphs, where developmental genes are expected to be more important than in adults.

*de novo* transcript evidence indicates that *D. citri hth* and *PKNOX* each have two isoforms (Table 3). In both cases, one isoform produces a protein with a MEIS domain [27] and the homeodomain (Dcitr03g04000.1.1 and MCOT13578.2.CT), while the other isoform encodes a protein with a MEIS domain but no homeodomain (Dcitr03g04000.1.2 and MCOT13578.1.CT). Homeodomain-less isoforms of Hth [66] and PKNOX2 [67] have also been reported in *Drosophila* and vertebrates, respectively. In *Drosophila*, the homeodomain-less forms of Hth can perform most of the protein's usual functions, but cannot specify antennal development [66].

## PBC genes

Exd is the insect member of the PBC class of Hox cofactors. The *D. citri* genome contains two apparent *exd* genes rather than one, as found in most other insects (Tables 2 and 3). While one of the genes has a typical eukaryotic gene structure with seven exons and six introns, the other gene is unusual because it contains no introns (Figure 6A). This gene structure suggests that the second *exd* gene is a retrogene, which forms by retrotransposition of a spliced mRNA [68]. The two *exd* genes encode proteins with 84% identity to one another. The scattered differences seen between the two proteins are consistent with gene duplication and divergence (Figure 6B). Moreover, the two genes map to separate chromosomes (*exd* on the X chromosome and exd-*r* on chromosome 1) and are flanked by different genes, strengthening the case for legitimate gene duplication. We have therefore named the intronless gene *extradenticle-retrogene* (*exd-r*) and the more typical gene *exd*.



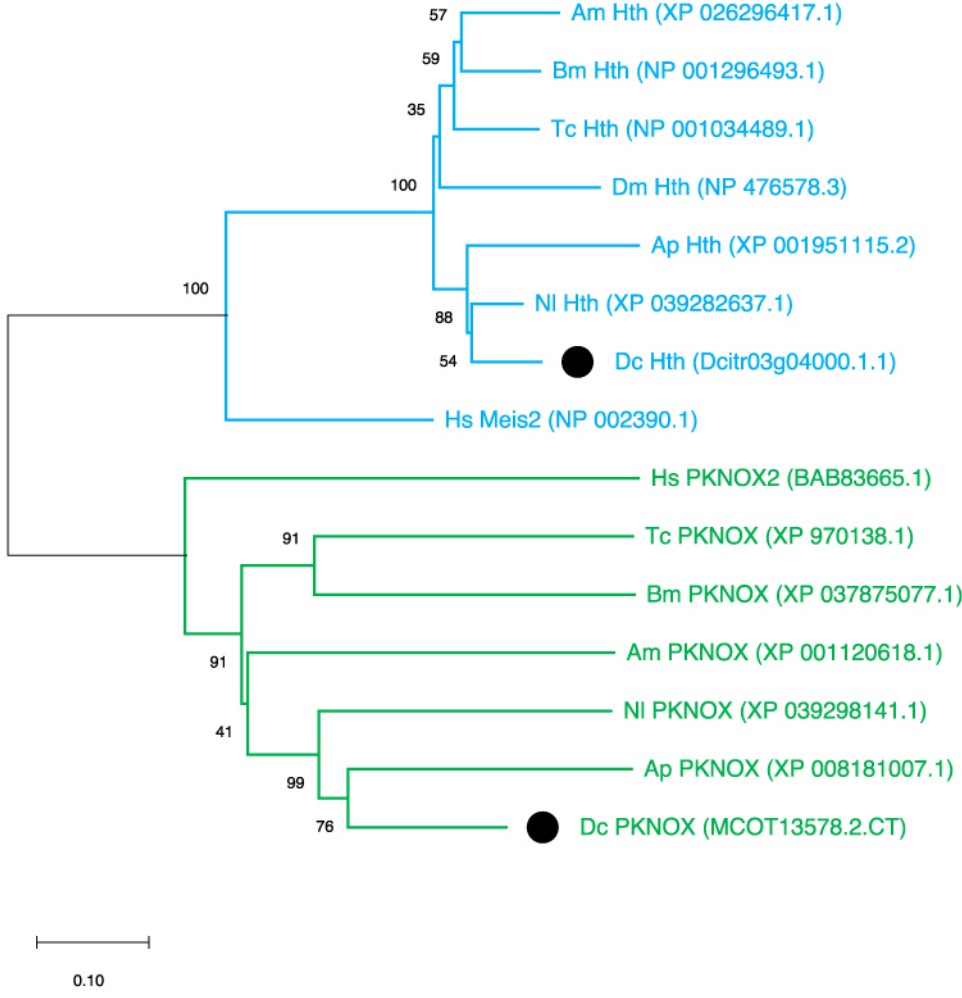

**Figure 4.** Phylogenetic tree of MEINOX family proteins. Full-length proteins were aligned with CLUSTALW in MEGAX [43]. The tree was constructed using the neighbor-joining method with 500 bootstrap replicates. Blue branches are part of the Hth/MEIS clade and green branches indicate the PKNOX/PREP clade. *Apis mellifera* (Am), *Bombyx mori* (Bm), *Tribolium castaneum* (Tc), *Drosophila melanogaster* (Dm), *Acyrthosiphon pisum* (Ap), *Nilaparvata lugens* (Nl), *Diaphorina citri* (Dc), *Homo sapiens* (Hs). *D. citri* proteins are denoted by a black circle. The scale bar for branch length is measured in amino acid substitutions per site.

There is a second copy of *exd-r* adjacent to the annotated locus, but it appears to be a false duplication resulting from incomplete collapse of haplotypes during genome assembly.

*exd-r* does not seem to have retained the 3′ poly(A) region (the vestige of a poly(A) tail) that is still present in some retrogenes [69]. This is not surprising, as a study of more than 20 young retrogenes in the *D. melanogaster* genome found that almost all had already lost the poly(A) tail [70]. At the 5′ end of *exd-r*, identity to *exd* at the nucleotide level starts at the translation start site. Such truncation is also common in retrogenes, possibly because of incomplete reverse transcription during the retrotransposition process [71].

Retrogenes do not usually include the regulatory elements of the parent gene, so they are not likely to retain their original expression pattern. This means that a retrogene will only be expressed if it is activated by regulatory elements close to its new position. If a retrogene is not expressed and therefore not subject to selection, it will often accumulate



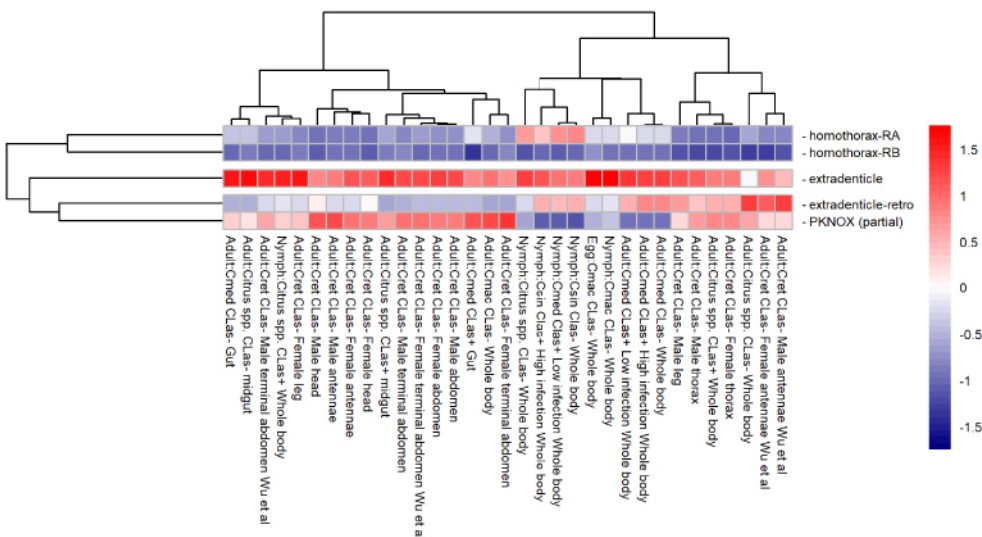

**Figure 5.** Expression of *exd*, *hth* and *PKNOX* genes in *Diaphorina citri* RNA-seq samples. Heatmap of relative expression levels of genes in RNA-seq samples from various developmental stages, tissues, host plants and *C*Las infection states ([53, 62–65] and NCBI BioProjects PRJNA448935 and PRJNA609978). Host plants are *Citrus macrophylla* (Cmac), *Citrus reticulata* (Cret), *Citrus* spp., *Citrus medica* (Cmed), and *Citrus sinensis* (Csin). The heatmap was scaled by column to allow comparison of expression of different genes within samples. *PKNOX* is expressed at higher levels than *hth-RA* in the adult samples, while *hth-RA* is expressed at higher levels than *PKNOX* in many of the nymph samples. Both *exd* and *exd-r* are expressed, raising the possibility that they could both be functional.

mutations and become a pseudogene. However, the *exd-r* open reading frame (ORF) seems to be complete, with no premature stop codons. Moreover, RNA-seq evidence from CGEN indicates that the *exd-r* locus is expressed at a low-to-moderate level in most tissues and developmental stages (Figure 5, Table 5). To confirm that the expression seen in CGEN was not the result of multimapped or mismapped reads actually produced by the original *exd* locus, we examined the reads mapping to *exd* or *exd-r* in Apollo. The mapped reads show very few mismatches with the genomic sequence at each locus, indicating that transcripts from both genes are present in the RNA-seq samples. Taken together, these observations suggest that *exd-r* has not become a pseudogene and could be a functional gene.

To determine if *exd-r* is evolutionarily conserved, we searched other hemipteran genomes for *exd* homologs. The closest sequenced relative of *D. citri* is the hackberry petiole gall psyllid *Pachypsylla venusta* [72]. We found orthologs of both Exd-r (98% identity) and Exd (98.5% identity) encoded in the *P. venusta* genome. However, we found no evidence of an *exd* retrogene in any other genomes, including those of three other sequenced members of the suborder Sternorrhyncha: the sweet potato whitefly, *B. tabaci* [73], the greenhouse whitefly, *Trialurodes vaporariorum* [74], and the pea aphid, *A. pisum* [75]. Thus, we can place the origin of the retrogene after the separation of the psyllid lineage from the other sternorrhynchans, but before the divergence of *D. citri* and *P. venusta*.

Surprisingly, when the two *D. citri* Exd proteins are compared with other hemipteran Exd proteins, Exd-r has a higher percent identity to the orthologs (>90%) than the "typical" *D. citri* Exd protein (80–83%). In fact, Exd-r has higher overall identity to its hemipteran orthologs than it does to the other Exd protein from *D. citri* (82.5%). Only at the extreme



**Table 5.** TPM expression values for Hox cofactor genes.

| Gene ID | Dcitr08g10400.1.1 | Dcitr01g03720.1.1 | Dcitr03g04000.1.1 | Dcitr03g04000.1.2 | Dcitr00g14970.1.1 |
|---|---|---|---|---|---|
| Gene/transcript symbol | *exd* | *exd-r* | *hth*-RA | *hth*-RB | *PKNOX* partial |
| Egg *C. macrophylla C*Las− Whole body | 96.9 | 33.25 | 32.58 | 16.98 | 23.56 |
| Nymph *C. medica C*Las+ Low infection Whole body | 9.44 | 7.38 | 8.61 | 0.09 | 0 |
| Nymph *C. sinensis C*Las+ High infection Whole body | 8.52 | 6.25 | 5.56 | 0.4 | 0 |
| Nymph *C. sinensis C*Las− Whole body | 11.06 | 9.25 | 10.7 | 0.44 | 0 |
| Nymph *C. macrophylla C*Las− Whole body | 37.17 | 10.95 | 10.35 | 0.64 | 7.51 |
| Nymph *Citrus* spp. *C*Las− Whole body | 21.9 | 8.24 | 16.46 | 0 | 5.43 |
| Nymph *Citrus* spp. *C*Las+ Whole body | 18.02 | 6.1 | 2.92 | 0.35 | 9.76 |
| Adult *C. medica C*Las− Gut | 9.97 | 2.13 | 2.53 | 0.29 | 5.14 |
| Adult *C. medica C*Las+ Gut | 5 | 2.26 | 2.88 | 0.21 | 5.72 |
| Adult *C. medica C*Las+ High infection Whole body | 12.6 | 9.73 | 3.99 | 0.12 | 0.04 |
| Adult *C. medica C*Las+ Low infection Whole body | 12.57 | 7.88 | 4.94 | 0.05 | 0 |
| Adult *C. medica C*Las− Whole body | 14.24 | 11.24 | 4.45 | 0 | 0 |
| Adult *C. macrophylla C*Las− Whole body | 12.1 | 4.08 | 4.31 | 1.49 | 13.61 |
| Adult *Citrus* spp. *C*Las− Whole body | 6.09 | 11.49 | 3.76 | 0.85 | 8.69 |
| Adult *Citrus* spp. *C*Las+ Whole body | 13.16 | 11.18 | 2.65 | 1.28 | 12.97 |
| Adult *Citrus* spp. *C*Las− midgut | 26.85 | 3.38 | 5.31 | 0 | 10.85 |
| Adult *Citrus* spp. *C*Las+ midgut | 17.99 | 2.12 | 2.41 | 0.2 | 12.13 |
| Adult *C. reticulata C*Las− Female abdomen | 28.77 | 6.47 | 2.73 | 0 | 23.42 |
| Adult *C. reticulata C*Las− Female antennae | 18.43 | 6.89 | 1.57 | 0.98 | 16.45 |
| Adult *C. reticulata C*Las− Female head | 19.89 | 10.41 | 1.3 | 0.26 | 18.13 |
| Adult *C. reticulata C*Las− Female leg | 38.67 | 10.51 | 1.76 | 0.91 | 20.14 |
| Adult *C. reticulata C*Las− Female terminal abdomen | 15.75 | 1.89 | 0.35 | 0 | 22.38 |
| Adult *C. reticulata C*Las− Female thorax | 32.42 | 27.35 | 4.7 | 2.56 | 31.49 |
| Adult *C. reticulata C*Las− Male abdomen | 19.28 | 4.77 | 2.27 | 0.27 | 16.38 |
| Adult *C. reticulata C*Las− Male antennae | 17.31 | 7.91 | 2.32 | 1.63 | 20.16 |
| Adult *C. reticulata C*Las− Male head | 19.26 | 12.43 | 2.41 | 0.6 | 22.9 |
| Adult *C. reticulata C*Las− Male leg | 27.96 | 21.68 | 3.79 | 0.47 | 17.48 |
| Adult *C. reticulata C*Las− Male terminal abdomen | 15.32 | 2.28 | 0.51 | 0.18 | 13.02 |
| Adult *C. reticulata C*Las− Male thorax | 33.75 | 23.23 | 5.12 | 1.17 | 27.09 |
| Adult *C. reticulata C*Las− Female antennae [60] | 22.51 | 27.08 | 5.95 | 0.41 | 17.26 |
| Adult *C. reticulata C*Las− Female terminal abdomen [60] | 20.19 | 5.46 | 3.47 | 0.52 | 17.48 |
| Adult *C. reticulata C*Las− Male antennae [60] | 27.65 | 42.67 | 7.37 | 0.85 | 24.39 |
| Adult *C. reticulata C*Las− Male terminal abdomen [60] | 16.47 | 5.2 | 2.34 | 0 | 10.66 |

Expression levels of Hox cofactor genes used to create the heatmap in Figure 5. Counts were obtained from publicly available datasets in CGEN [40] and are reported in transcripts per million (TPM). Developmental stage, citrus host, *C*Las infection status and source tissue for each sample are noted in the first column.

N-terminus does *D. citri* Exd-r appear more similar to *D. citri* Exd than to Exd proteins from other hemipterans (Figure 6B). Phylogenetic analysis is consistent with these observations. The *D. citri* and *P. venusta* Exd-r proteins cluster with other hemipteran Exd proteins, while their Exd proteins cluster as an outgroup to the insect Exd cluster (Figure 7).

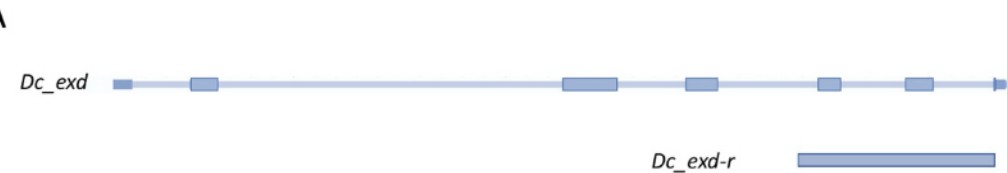

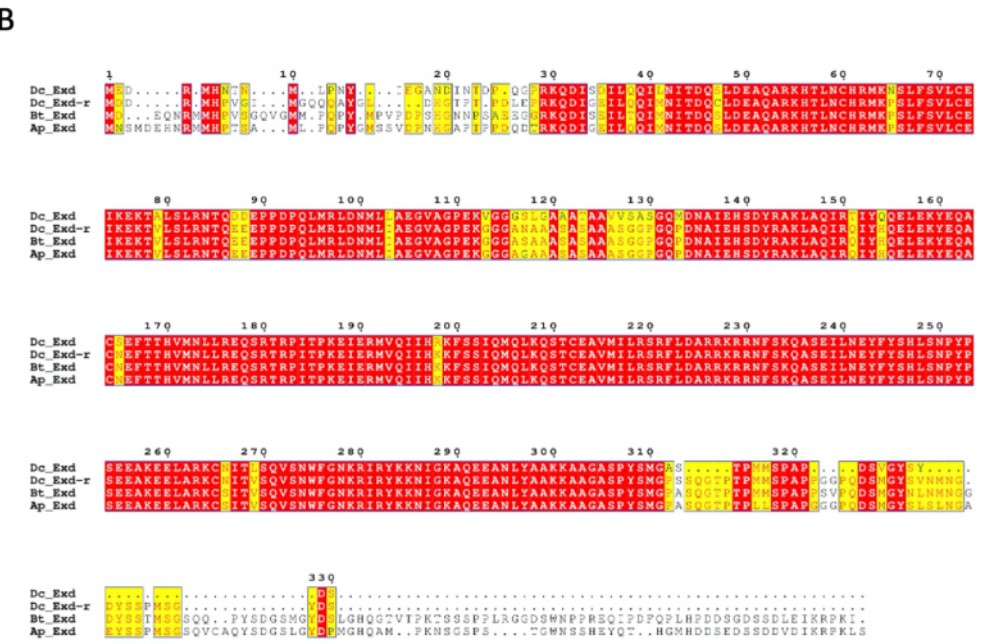

**Figure 6.** Comparison of the two *extradenticle* (*exd*) genes in *Diaphorina citri*. (A) Structural comparison of the *D. citri exd* and *extradenticle-retro* (*exd-r*) transcripts. *exd* has 7 exons and six introns, while *exd-r* has a single long exon with no introns. (B) Multiple sequence alignment of Exd proteins from three hemipteran species: *D. citri* Exd (Dc_Exd), Exd-r (Dc_Exd-r), *Bemisia tabaci* Exd (Bt_Exd) and *Acyrthosiphon pisum* Exd (Ap_Exd). Residues identical in all four proteins are shaded red, while those identical in three out of four are shaded yellow. At yellow-shaded residues, *D. citri* Exd-r often matches the orthologs in other species rather than its *D. citri* paralog Exd.

The high level of conservation between Exd-r and the Exd proteins of other species supports the possibility that Exd-r still has a function. Functional retrogenes are not unheard of; the *Drosophila* genome has almost 100 retrogenes with apparent functions [54]. It is more difficult to explain the observation that Exd-r more closely resembles its hemipteran orthologs than it does its paralog Exd, since the paralogs diverged more recently, so they would be expected to be more closely related. It appears that Exd, rather than Exd-r, as might have been expected, has diverged significantly since the duplication event, resulting in decreased identity to both Exd-r and their orthologs. Since the *D. citri* and *P. venusta* Exd sequences are almost identical, most of the divergence must have occurred before separation of these psyllid lineages. One explanation for this scenario is that *exd-r* inserted in a location that allowed it to maintain its original expression pattern and function, freeing *exd* to diverge. While it is usually the retrogene that acquires a new function after duplication, there is precedent for the parental gene diverging in expression pattern and function. For example, the *Drosophila* retrogene *e(y)2* has replaced the function of its parental gene *e(y)2b* [55]. The lack of additional divergence between the *D. citri* and

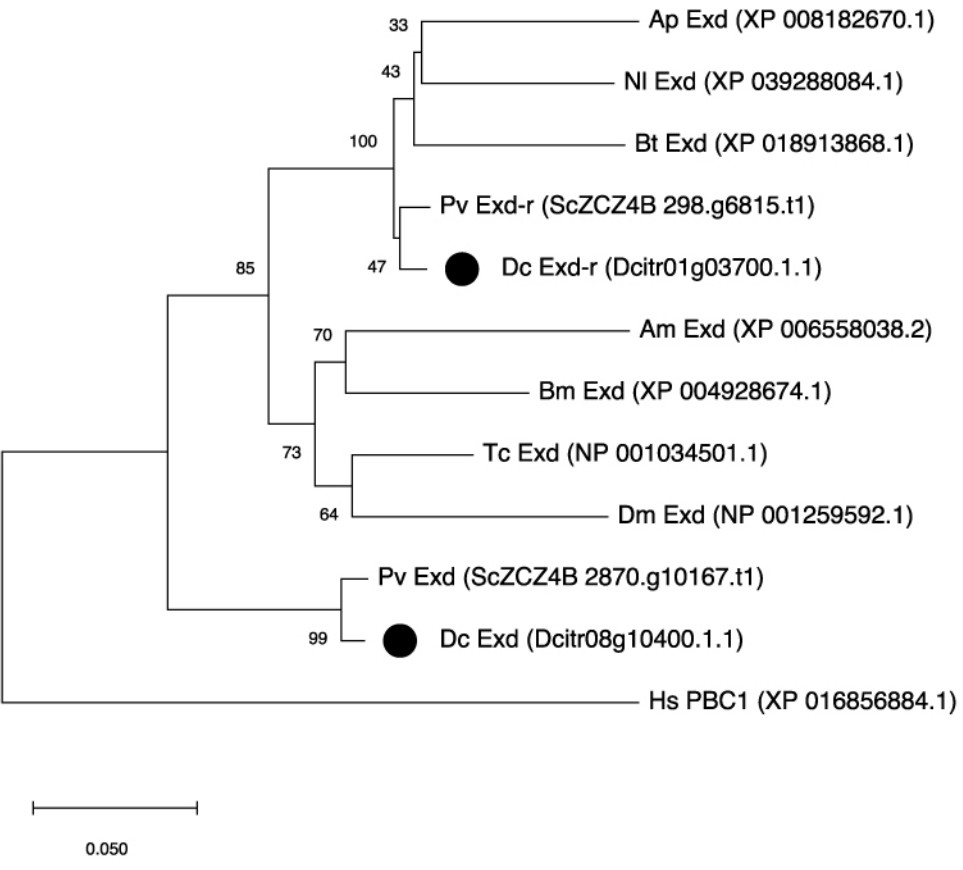

**Figure 7.** Phylogenetic tree of Exd/PBC family proteins. Full-length proteins were aligned using CLUSTALW within MEGAX [43]. A neighbor-joining tree was constructed with 500 bootstrap replicates. *Apis mellifera* (Am), *Bombyx mori* (Bm), *Tribolium castaneum* (Tc), *Drosophila melanogaster* (Dm), *Acyrthosiphon pisum* (Ap), *Nilaparvata lugens* (Nl), *Bemisia tabaci* (Bt), *Pachypsylla venusta* (Pv), *Diaphorina citri* (Dc), *Homo sapiens* (Hs). *Diaphorina citri* proteins are marked by a black circle. The scale bar refers to amino acid substitutions per site.

*P. venusta* Exd proteins is intriguing and suggests that Exd may have acquired a new function in the ancestral psyllid that constrained further sequence change. Functional studies of both Exd and Exd-r will be needed to test this hypothesis.

## RE-USE POTENTIAL

*D. citri* is the focus of a major pest control effort because of its economic impact on the citrus industry as the vector of citrus greening disease. Gene-based control methods are desirable, since pesticides are not a sustainable solution. Our improved annotations will allow researchers to design more accurate, detailed experiments on genes of interest. Our revised gene models will be part of a new official gene set that will be available for BLAST analysis and expression profiling on the Citrus Greening website and the CGEN [44].

## DATA AVAILABILITY

The *D. citri* genome assembly, gene sets, and transcriptomic data can be accessed through the Citrus Greening website [44]. Accession numbers for all genes used in phylogenetic

analysis are provided within this report, and all additional data are available via the *GigaScience* GigaDB repository [56].

## EDITOR'S NOTE
This article is one of a series of Data Releases crediting the outputs of a student-focused and community-driven manual annotation project curating gene models and, if required, correcting assembly anomalies, for *the Diaphorina citri* genome project [57].

## DECLARATIONS
## LIST OF ABBREVIATIONS
*abd-A*: *abdominal-A*; *Abd-B*: *Abdominal-B*; Am: *Apis mellifera*; Antp: *Antennapedia*; Ap: *Acyrthosiphon pisum*; Bm: *Bombyx mori*; Bt: *Bemisia tabaci*; CGEN: Citrus Greening Expression Network; *C*Las: *Candidatus* Liberibacter asiaticus; Dc: *Diaphorina citri*; *Dfd*: *Deformed*; Dm: *Drosophila melanogaster*; *exd*: *extradenticle*; *exd-r*: *extradenticle-retrogene*; *ftz*: *fushi tarazu*; Hs: *Homo sapiens*; *hth*: *homothorax*; *lab*: *labial*; miRNA: microRNA; MCOT: Maker, Cufflinks, Oases, Trinity; NCBI: National Center for Biotechnology Information; Nl: *Nilaparvata lugens*; *pb*: *proboscipedia*; Pv: *Pachypsylla venusta*; *Scr*: *Sex combs reduced*; Tc: *Tribolium castaneum*; TPM: transcripts per million; Ubx: *Ultrabithorax*; *zen*: *zerknüllt*.

## ETHICAL APPROVAL
Not applicable.

## CONSENT FOR PUBLICATION
Not applicable.

## COMPETING INTERESTS
The authors declare that they have no competing interests.

## FUNDING
This research was funded by USDA-NIFA grant 2015-70016-23028, HSI 1300394, 2020-70029-33199 and an Institutional Development Award (IDeA) from the National Institute of General Medical Sciences of the National Institutes of Health under grant number P20GM103418.

## AUTHOR CONTRIBUTIONS
WBH, SJB, TD and LAM conceptualized the study; TD, SS, TDS and SJB supervised the study; SJB, TD, SS, and LAM contributed to project administration; TDS conducted the investigation; PH, MF-G, and SS contributed to software development; SS, TDS, PH, and MF-G developed methodology; SJB, TD, WBH, and LAM acquired funding; TDS prepared and wrote the original draft; SS, WBH, TD and SJB reviewed and edited the draft.

## ACKNOWLEDGEMENTS
We thank Dr. Josh Benoit for assistance with using pheatmap to visualize expression data.

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
