## [Reviewer Report]

Comments on revised manuscriptThe authors have appropriately addressed my concerns，and I do not have further questions.

---

## [Reviewer Report]

Reviewer name and names of any other individual's who aided in reviewer HailinDo you understand and agree to our policy of having open and named reviews, and having your review included with the published papers. (If no, please inform the editor that you cannot review this manuscript.)YesIs the language of sufficient quality?YesPlease add additional comments on language quality to clarify if needed
Are all data available and do they match the descriptions in the paper? NoAdditional CommentsAre the data and metadata consistent with relevant minimum information or reporting standards? See GigaDB checklists for examples <a href="http://gigadb.org/site/guide" target="_blank">http://gigadb.org/site/guide</a>YesAdditional CommentsIs the data acquisition clear, complete and methodologically sound?NoAdditional CommentsIs there sufficient detail in the methods and data-processing steps to allow reproduction?YesAdditional CommentsIs there sufficient data validation and statistical analyses of data quality? YesAdditional CommentsIs the validation suitable for this type of data?YesAdditional CommentsIs there sufficient information for others to reuse this dataset or integrate it with other data?YesAdditional CommentsAny Additional Overall Comments to the AuthorMinor revision please:
1. This manuscript need to be reorganized, as the method, result and discussion were somewhat mixed.
2. Line 125, were these data newly got? How much data you used should also be presented.
3. How do you make sure that the hox genes you find were complete or exact? Was there any validation?RecommendationMinor Revision

---

## [Reviewer Report]

Reviewer name and names of any other individual's who aided in reviewer Mary Ann TuliDo you understand and agree to our policy of having open and named reviews, and having your review included with the published papers. (If no, please inform the editor that you cannot review this manuscript.)YesIs the language of sufficient quality?YesPlease add additional comments on language quality to clarify if needed
NAAre all data available and do they match the descriptions in the paper? YesAdditional CommentsThe author states, "Reciprocal BLAST was used to confirm orthologs for all D. citri genes", and has explained (through the pre-review process) that these were performed manually on the NCBI website over a period of months by different authors and thus cannot be easily reproduced. I think it could be made more clear that this is in line with manual curation and the accession numbers are all provided in the paper.Are the data and metadata consistent with relevant minimum information or reporting standards? See GigaDB checklists for examples <a href="http://gigadb.org/site/guide" target="_blank">http://gigadb.org/site/guide</a>YesAdditional CommentsIs the data acquisition clear, complete and methodologically sound?YesAdditional CommentsIs there sufficient detail in the methods and data-processing steps to allow reproduction?YesAdditional CommentsSee above comment regarding reciprocal BLASTIs there sufficient data validation and statistical analyses of data quality? YesAdditional CommentsIs the validation suitable for this type of data?YesAdditional CommentsIs there sufficient information for others to reuse this dataset or integrate it with other data?YesAdditional CommentsIt does meet reuse criteria, but will be more reusable once the data is available from the Citrus Greening website.Any Additional Overall Comments to the AuthorRecommendationMinor Revision